# Policy Optimization via Stochastic Recursive Gradient Algorithm

## Abstract

In this paper, we propose the StochAstic Recursive grAdient Policy Optimization (SARAPO) algorithm which is a novel variance reduction method on Trust Region Policy Optimization (TRPO). The algorithm incorporates the StochAstic Recursive grAdient algoritHm(SARAH) into the TRPO framework. Compared with the existing Stochastic Variance Reduced Policy Optimization (SVRPO), our algorithm is more stable in the variance. Furthermore, by theoretical analysis the ordinary differential equation and the stochastic differential equation (ODE/SDE) of SARAH, we analyze its convergence property and stability. Our experiments demonstrate its performance on a variety of benchmark tasks. We show that our algorithm gets better improvement in each iteration and matches or even outperforms SVRPO and TRPO.

## 1 Introduction

Reinforcement Learning (RL) (Sutton et al., 1998) is a dynamic learning approach to interact with the environment and make actions so certain measure of cumulative rewards is maximized. It achieves remarkable performance in several tasks including continuous control, video games, etc. Among all existing RL algorithms, policy gradient (Sutton et al., 1999) is the most fundamental method for RL.

However, policy gradient suffers from sample inefficiency and high variance during the training phase. We could use reward shaping techniques such as, REINFORCE(Williams, 1992), GAE(Schulman et al., 2015b), etc., as an variance reduction. Besides, trust region policy optimization (TRPO) was proposed by (Schulman et al., 2015a) to remedy this problem. It merges the trust region method (Nocedal & Wright, 2006) and the natural gradient theory (Amari, 1998) into the policy gradient framework. The basic idea is to make the policy move toward the direction that improves mean episode rewards under the constraint of the Kullback-Leibler (KL) divergence with the old policy distribution.

Compared to traditional gradient descent which corresponds to the $L^2$ constraint, constraints that uses KL distance has been widely adopted as an alternative choice. The KL constraint results a natural gradient update firstly proposed by Amari (1998). Later Kakade (2001) merges the natural gradient into the policy gradient (See Martens (2014) for further extensions). Natural gradient attempts to solve the problem of vanishing gradients in the so-called plateau areas. TRPO which strictly constrain the KL distance achieves a more robust performance on continuous control tasks.

Policy gradient optimization method shares similar structure with traditional optimization thus can be powerfully augmented by variance reduced gradient methods. For example, SVRG (Johnson & Zhang, 2013) is a kind of stochastic variance reduced method that overcomes the fallback of SGD of large variance and potentially accelerates the RL training. Previous works use SVRG to accelerate policy gradient (SVRPG) and its TRPO variants (SVRPO) (Xu et al., 2017; Papini et al., 2018). The variance reduced version of policy gradient allows a multi-step optimization for each batch of data, while the vanilla policy gradient optimizes the policy once each iteration. In theory, SVRG accommodates with larger stepsize for each iteration and observably outperforms traditional PG and TRPO, respectively. Furthermore, the introduction of stochastic variance reduction not only improves sample efficiency but also stabilizes the training process by reducing the variance and accommodate larger stepsize.

In this paper, we focus on studying variance reduction method on TRPO. We discussed an alternative variance reduction method called SARAH, recently proposed by (Nguyen et al., 2017) whose convergence rate matches that of SVRG in the convex case and has the inner-loop convergence property that SVRG does not possess. We propose a new algorithm named SARAPO which hybrids the SARAH method with TRPO and analyze the dynamics of SVRG/SARAH on natural gradient updates via its approximating ordinary and stochastic differential equation (ODE/SDE).

In experiments, we test the performance and sample efficiency of SARAPO, SVRPO, and TRPO. To allow larger steps in SVRPO/SARAPO, we remove the KL constraint in the line search of TRPO. We verify that variance reduction enables a larger leap towards the optima and find that SARAPO outperforms TRPO and matches the performance of SVRPO in most of the tasks.

The rest of this paper is organized as the following: we introduce some background, propose and analyze our algorithm in Section 2. In Section 3, we theoretically analyze our proposed SARAPO algorithm from a dynamical system viewpoint. We present the numerical results of experiments in Section 4 and conclude our paper in Section 5.

## 2 PRELIMINARIES AND METHODOLOGY

The reinforcement learning task can be considered as solving a discrete time Markov Decision Process (MDP) $M = \{\mathcal{S}, \mathcal{A}, \mathcal{P}, \mathcal{R}, \gamma, \rho\}$, where $\mathcal{S}$ is the set of states, $\mathcal{A}$ is the set of actions, $\mathcal{R} : \mathcal{S} \times \mathcal{A} \rightarrow \mathbb{R}$ is the reward function, $\gamma$ is the discount factor and $\rho$ the initial state distribution. The policy gradient method take steps by directly maximizing the expected sum of discounted rewards:

$$L(\pi_w) = \mathbb{E}_{\pi_w} \left[ \sum_{t=0}^{\infty} \gamma^t r_t(s_t, a_t) \right]$$

with respect to the parameters $w$ of the stochastic policy $\pi_w(a|s)$. The gradient of the objective function $\nabla_w L$ is given by:

$$\nabla_w L(w) = \mathbb{E}_{\pi_w} \left[ \sum_{t=0}^{\infty} \nabla_w \log \pi(a_t|s_t) Q^\pi(s_t, a_t) \right], \tag{1}$$

where $Q^\pi$ is the state-action function (i.e., the Q function) defined as the expected return when taking action $a_t$ from state $s_t$. The expectatioin is calculated by Monte-Carlo simulation by sampling $n$ timesteps $\{s_t, a_t, r_t\}_{t=1}^n$ of policy $\pi_w$.

### 2.1 TRUST REGION POLICY GRADIENT

As the policy gradient is highly unstable and sample inefficient, TRPO introduces the trust region method to policy gradient to limit how far the new policy is deviated from the old policy. Instead of the $L^2$ norm in trust region method, TRPO takes KL as the measure of distance. Furthermore, to make computation easy, the algorithm turns the KL constraint problem into a penalized problem. TRPO optimizes the surrogate loss at every iteration:

$$\max_w \quad L_{w_{old}}(w) = \mathbb{E}_{\pi_{w_{old}}} \left[ \frac{\pi_w(a|s)}{\pi_{w_{old}}(a|s)} A^{\pi_{w_{old}}}(s, a) \right] - \beta \overline{D}_{KL}(\pi_{w_{old}}, \pi_w) \tag{2}$$

At every iteration, TRPO takes an update of:

$$w_{t+1} = w_t + \eta_t F^{-1} v_t \tag{3}$$

where $v_t = \nabla_w L$ and $F$ is the Fisher Information Matrix (FIM)

$$F = E_{a,s \sim pi_w}[\nabla \log \pi_w(a|s) \cdot \nabla \log \pi_w(a|s)^T].$$

The initial step size $\eta$ is calculated by limiting the KL distance within the range $\delta$:

$$\eta_t = \sqrt{2\delta/(v_t F^{-1} v_t)} \tag{4}$$

---

**Algorithm 1** SARAPO

---

  **Initialization:**
Initialize $w_0 = w_1$ to be a randomized parameter.
**for** $t = 1$ to $T$ **do**
    **if** $t\%m == 0$ **then**
        Set $\tilde{w} = w_t$
        Run policy $\pi_{w_t}$ for N timesteps. Store a batch of trajectories $\mathcal{D}$. Calculate the baseline
gradients:

$$\tilde{v}_0 = \frac{1}{|\mathcal{D}|} \sum_{x \in \mathcal{D}} \nabla L_x(w_t)$$

    **else**
        Draw a mini-batch $\mathcal{D}_t$ of size $M$ from $\mathcal{D}$
        Calculate the estimated gradient $w_t$:

$$\tilde{v}_t = \tilde{v}_{t-1} + \frac{1}{|\mathcal{D}_t|} \sum_{x \in \mathcal{D}_t} \left( \nabla L_x(w_t) - \nabla L_x(w_{t-1}) \right)$$

    **end if**
    Update policy parameter with Algorithm 2
**end for**
**Ensure:** $w \leftarrow \tilde{w}^T$

---

## 2.2 SARAH AND SVRG FOR NATURAL GRADIENT

SARAH (Nguyen et al., 2017) and SVRG (Johnson & Zhang, 2013) are among the recently popularized variance reduced gradient algorithms in optimization. They reduce the variance during the optimization process and achieve a faster convergence rate. Instead of calculating the full gradient $\nabla L$.

Recall that SVRG calculate a mini batch of size $m$ in each inner loop, and it updates the weights as follows:

$$v_{t+1} = \tilde{v} + \nabla_w L_m(w_t) - \nabla_w L_m(\tilde{w})$$
$$w_{t+1} = w_t - \eta F^{-1} v_{t+1}$$

where $\tilde{v}$ and $\tilde{w}$ stores the full gradient and weight parameters as the output of the previous loop, and $L_m$ is the objective function estimated using $m$ samples. Here the update rule of $w_t$ involves the inverse of the Fisher information matrix as in TRPO, which is computed via CG method. Also, $v_t$ provides an unbiased estimation of the true gradient while the variance is much reduced when $w_t$ is close to $\tilde{w}$

SARAH, on the other hand, estimates each gradients using a recursive gradient update rule. It adds the gradient estimated one step before $v_t$ by the difference of gradients between the current parameter and the previous parameter $\nabla_w L_m(w_t) - \nabla_w L_m(w_{t-1})$ estimated over a mini batch. The core update rule is as follows:

$$v_{t+1} = v_t + \nabla_w L_m(w_t) - \nabla_w L_m(w_{t-1})$$
$$w_{t+1} = w_t - \eta F^{-1} v_{t+1}$$

In SARAH, the algorithm exchanges unbiasedness of the gradient with a much reduced variance of such. We will analyze the iteration in Section 3.

## 2.3 STOCHASTIC GRADIENT RECURSIVE POLICY OPTIMIZATION

Our goal in this section is to combine the SARAH with TRPO for natural gradient and formally propose the SARAPO algorithm. The details of calculating the estimated gradients are listed in Algorithm 1. There, we use $\tilde{w}$ to denote the old policy parameters saved at the beginning of each outer loop. After calculating the estimate of the gradient, we conduct a TRPO step. We note that as the Fisher Information Matrix can be approximated by Hessian matrix of the KL divergence when

---

**Algorithm 2** Inner loop details

---

Estimate the Fisher Information Matrix $\hat{F} = \frac{\partial^2}{\partial w_n \partial w_m} \overline{KL}(\pi_{w_t}(\cdot \| s_n), \pi_w(\cdot \| s_n))|_{w_t}$
Update policy parameter with mini-batch $I_j$:

$$w_{t+1} \leftarrow w_t + \eta_t \hat{F}^{-1}(w_t) \cdot \hat{v}_t.$$

where the $\hat{F}^{-1}(w_t) \cdot \hat{v}_t$ is computed using the conjugate gradient method
Initialize $\eta_t$ according to equation 4
Line-search $\eta_j, \eta_j/2, \eta_j/4, \cdots$ for improving $L_{\pi_{\tilde{w}}(\pi_w)}$

---

the current distribution exactly matches that of the base distribution. As a consequence, we should use the second derivative of the KL distance between the current parameters and the parameters one step before to estimate $\hat{F}$, which is to penalize the KL between each inner loop updates, instead of the outer loop updates in TRPO. The Fisher information matrix is only estimated using a mini batch of the data. And the inverse Fisher vector product is calculated by the conjugate gradients algorithm.

Different from the line-search process in TRPO, we do line-search only to find the stepsize $\eta$ that can improve the policy loss. Due to the fact that the variance is reduced, we can take a larger step away. So we remove the KL condition in the line-search and let the algorithm to find the KL distance of the outer loop updates itself. Detailed description is in Sec 4.

## 3 THEORY OF DYNAMICAL SYSTEMS

Recall that the core inner-loop update rule of Stochastic Recursive Gradient Method (SARAH) (Nguyen et al., 2017) is, assuming the minibatch size is 1,

$$
\begin{aligned}
v_{t+1} &= v_t + \nabla L_{i_{t+1}}(w_t) - \nabla L_{i_{t+1}}(w_{t-1}), \\
w_{t+1} &= w_t - \eta v_{t+1}.
\end{aligned}
$$

Here we provide a simple alternative viewpoint of (random) dynamical systems. The method of using dynamical systems and continuous-time framework have been adopted to characterize variants of gradient descent algorithms (Su et al., 2016; Li et al., 2017), but to our best knowledge, it has never been used to analyze variance reduced gradient methods before this work.

As we could see from Algorithm 1, we applied a *scaled-form* of SARAH in combination with Trust Region Policy Optimization by the inverse Fisher Information matrix $F^{-1}$. Seeing this, the inner loop update has the essential form of

$$
\begin{aligned}
v_{t+1} &= v_t + \nabla L_{i_{t+1}}(w_t) - \nabla L_{i_{t+1}}(w_{t-1}), \\
w_{t+1} &= w_t - \eta F^{-1} v_{t+1}.
\end{aligned}
\tag{5}
$$

For simplicity of analysis, we consider the *quadratic approximation* approach. Let the quadratic objective function $L(w) := \frac{1}{n} \sum_{i=1}^n L_i(w)$, with

$$L_i(w) = \frac{1}{2} w^\top H_i w + g_i^\top w.$$

We sketch the analysis using ODE and SDE, in order to better understand the dynamics.

### 3.1 ODE ANALYSIS

By rescaling the time $t = \eta^{-1}s$ according to the stepsize $\eta$, we define a new process

$$V(s) = v_{\lfloor \eta^{-1}s \rfloor} \quad \text{and} \quad X(s) = w_{\lfloor \eta^{-1}s \rfloor}.$$

Then the newly defined process is converted from equation 5 and has the form

$$
\begin{aligned}
V(s + \eta) &= V(s) + \nabla L_{i_{s+\eta}}(X(s)) - \nabla L_{i_{s+\eta}}(X(s - \eta)) \\
&= V(s) + HX(s) - HX(s - \eta) + \eta e_{s+\eta},
\end{aligned}
$$

where $e_{s+\eta}$ is a martingale difference noise term of $O(1)$, and

$$X(s + \eta) = X(s) - \eta F^{-1}V(s + \eta).$$

Substituting $X(s) - X(s - \eta)$ with $-\eta F^{-1}V(s)$, we derive the update rule for $V(s)$

$$\eta^{-1}(V(s + \eta) - V(s)) = -HF^{-1}V(s) + e_{s+\eta}.$$

From the standard weak convergence theory, one easily conclude the following:

**Theorem 1** *When $\eta \to 0^+$, the scaled SARAH update rule (5) can be approximated by the following ODE system*

$$
\begin{aligned}
\frac{d}{ds}V(s) &= -HF^{-1}V(s), \\
\frac{d}{ds}X(s) &= F^{-1}V(s).
\end{aligned}
\tag{6}
$$

*Note in (6), the dynamics of $V(s)$ forms an autonomous ODE, which is independent of the dynamics of $X(s)$.*

Left multiplying both sides of the first line of equation 6 by $F^{-1}$, one obtains

$$\frac{d}{ds}(F^{-1}V(s)) = -F^{-1}H(F^{-1}V(s)).$$

Noticing $V(0) = HX_0$, we can easily verify that the $V(s)$ can be solved as

$$V(s) = F(F^{-1}V(s)) = F\exp(-s \cdot F^{-1}H)F^{-1}HX_0 = H\exp(-s \cdot F^{-1}H)X_0. \tag{7}$$

By combining $V(s)$'s solution with the update rule for $X(s)$, we have

$$\frac{d}{ds}X(s) = -\exp(-s \cdot F^{-1}H)F^{-1}HX_0,$$

and has the expressed solution

$$X(s) = X_0 - (I - \exp(-s \cdot F^{-1}H))X_0.$$

As a result, we reach the following solution to (6):

$$X(s) = \exp(-s \cdot F^{-1}H)X_0. \tag{8}$$

From (7) and (8), both the vectors $X(s)$ and $V(s)$ exponentially decays with rate matrix $F^{-1}H$ which is consistent with the theory of natural gradient ascent (Kakade, 2001).

## 3.2 SDE ANALYSIS

The ODE analysis simply neglects the effect of noise. Here we turns to develop a stochastic differential equation approximation tool to exploit the effect of noise. Denote $H_i = H + E_i$, $H$ being the Hessian for $L(w)$. Further assume that the noise term $E_i$ is $\sigma diag(\chi^1, \dots, \chi^d)$ where $\chi^i$ are i.i.d. standard normal, for simplicity.

Define the coordinate-wise product of two vectors $(x_1, \dots, x_d)^\top \bullet (y_1, \dots, y_d)^\top = (x_1 y_1, \dots, x_d y_d)^\top$. Let $W(s) = (W_1(s), \dots, W_d(s))^\top$ be a $d$-dimensional standard Wiener processes. Then under standard assumptions, the inner loop has an update rule for $V(s)$ which can be written as

$$
\begin{aligned}
V(s + \eta) - V(s) &= -\eta HF^{-1}V(s) - \eta \sigma \, \chi \bullet (F^{-1}V(s)) \\
&= -\eta HF^{-1}V(s) - \eta^{1/2}\sigma \, (W(s + \eta) - W(s)) \bullet (F^{-1}V(s)).
\end{aligned}
$$

As $\eta \to 0^+$, it becomes a SDE system

$$\frac{d}{ds}V(s) = -HF^{-1}V(s) - \eta^{1/2}\sigma \left(\frac{d}{ds}W(s)\right) \bullet (F^{-1}V(s)),$$

which can be simplified as

$$dV(s) = -HF^{-1}V(s)ds - \eta^{1/2}\sigma(F^{-1}V(s)) \bullet dW(s). \qquad (9)$$

From the above analysis, one can verify that the approximating SDE in (9) introduces a small (size of $O(\eta^{1/2})$) noise deviation from its ODE curve, and hence enjoys similar property as the natural gradient descent update. In the case where $H$ is convex, the dynamics validates the property of inner-loop convergence in (Nguyen et al., 2017) [Theorem 1a & 1b], and hence the multi-loop geometric convergence [Theorem 2]. The per-step cost, however, turns to be evaluations of two individual gradients instead of the whole-batched gradients.

**Remark 1** *One may also conduct a similar analysis for SVRG and conclude the same approximating ODE for scaled SVRG. The approximating SDE, nevertheless, shall differ by an extra noise term that is proportional to the distance between the current iterate and the control point. This well-explains the inner-loop convergence and stability properties that SARAH enjoy upon SVRG.*

## 4 EXPERIMENTS

In this section, we design a set of experiments to investigate the following questions:

- Does the performance of SARAPO matches the performance of SVRPO or even better? Which algorithm trains faster and achieves better mean episode reward?
- What happens in the inner loop? Does the variance reduction methods reaches a better optimum than gradient descent?
- What is the actual KL divergence between the updated policy and the old policy?

Our implementation is based on the $modular\_rl$. We choose the commonly used MuJoCo (Todorov et al., 2012) environment on several tasks: Swimmer, Half-Cheetah, Walker, and Hopper, and we also test the algorithms on two classic continuous control tasks: Pendulum and CartPole. We compare our implementation of SARAPO with SVRPO and the original TRPO. The result is shown in Figure 1.

### 4.1 EXPERIMENTAL SETTINGS

We follow the settings of Xu et al. (2017) and Schulman et al. (2015a) to fix part of our parameters. We set the number of steps per iteration to be 50000 and the discount factor to be 0.995. We use a gaussian policy with a diagonal covariance matrix, whose mean is parameterized by a multi-layer perceptron (MLP). For all tasks, we use two hidden layers of size $(64, 64)$ with tanh activation function. We fine-tuned other parameters as follows: minibatch size chosen in $\{1000, 5000\}$, the number of inner loop iteration in $\{20, 50\}$, inner loop max Kullback-Leibler divergence being in $[0.005, 0.01]$, and the Conjugate Gradient damping factor is $0.1$. We list the detailed parameters of each experiment in the Appendix.

For comparing fairly, we chose five different random seeds and plotted our result as the average of the five runs. As the number of the outer loop iteration is fixed, the $x$-axis is proportional to the number of samples we have drawn and the curve corresponds to the sample efficiency.

### 4.2 COMPARISON BETWEEN SARAPO, SVRPO AND TRPO

In Figure 1, we compared the learning curve of three algorithms which are SARAPO, SVRPO and TRPO. On the classic continuous control tasks such as Pendunlum and CartPole which are 2D balancing tasks, SARAPO outperforms other two algorithms. Specifically, SARAPO converges faster than SVRPO and TRPO and gets high mean episode reward at the end stage of the training process. We see that in Half-Cheetah environment which is a 3D continuous-control locomotion task, SARAPO observably performs better than other two algorithms and achieves a high score in 500 epochs. In Walker and Hopper (3D continuous-control locomotion task), the three algorithms exhibits comparable performance. SARAPO converges faster in Swimmer but SVRPO reaches a better score.

We make two more tables to get a clearer view of our result. Table 1 presents the number of iterations needed to cross the threshold of 90 percent of the best score. In four of the tasks, SARAPO reaches

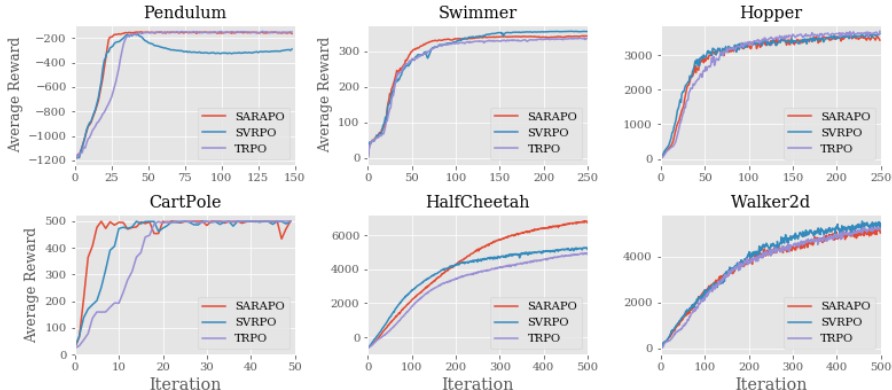

Figure 1: A comparison of variance reduced policy optimization and traditional TRPO. The result is the mean of five runs on different random seeds.

| Tasks | Threshold | SARAPO | SVRPO | TRPO |
|---|---|---|---|---|
| Pendulum | −250 | **23** | 27 | 34 |
| CartPole | 450 | **5** | 10 | 18 |
| Swimmer | 302 | **52** | 72 | 71 |
| Half-Cheetch | 4561 | **216** | 251 | 402 |
| Walker | 4669 | 343 | **274** | 330 |
| Hopper | 3198 | 74 | **65** | 78 |

Table 1: Number of Iteration Below the Threshold

| Tasks | SARAPO | SVRPO | TRPO |
|---|---|---|---|
| Pendulum | −149.4 | −167.9 | **−147.1** |
| CartPole | **500** | **500** | **500** |
| Swimmer | 343.0 | **355.7** | 336.5 |
| Half-Cheetch | **6845.1** | 5291.4 | 5080.9 |
| Walker | 5232.9 | **5529.2** | 5265.2 |
| Hopper | 3578.1 | 3625.8 | **3692.8** |

Table 2: Performance of Policy

the threshold the fastest and in the other two tasks, SVRPO outperforms the other two algorithms. In Table 2, on HalfCheetah task, SARAPO outperforms other two algorithms.

Table 2 shows the final performance of policy which is the average of best reward over five different seeds.

### 4.3 DETAILS

We perform further experiments to investigate the phenomenon during the training process. By analyzing on statistics of the small step updates and the outer loop updates respectively, we summarize three meaningful statistics: The norm of estimated gradients in each small step gives us a glimpse on how the gradient value is changing. The policy loss improvement is the absolute difference between the objective function value before and after an outer loop update, which intuitively visualize the expected increase in the mean episode reward. And the KL divergence between outer loop updates illustrates the difference between the old policy and the updated policy. Larger KL divergence and loss improvement are indicators of a good policy update.

Figure 4.3 incorporates the result of all three statistics of three different tasks. The first row shows the training process of Swimmer, the second row is the Hopper task and the third row is Half-Cheetah.

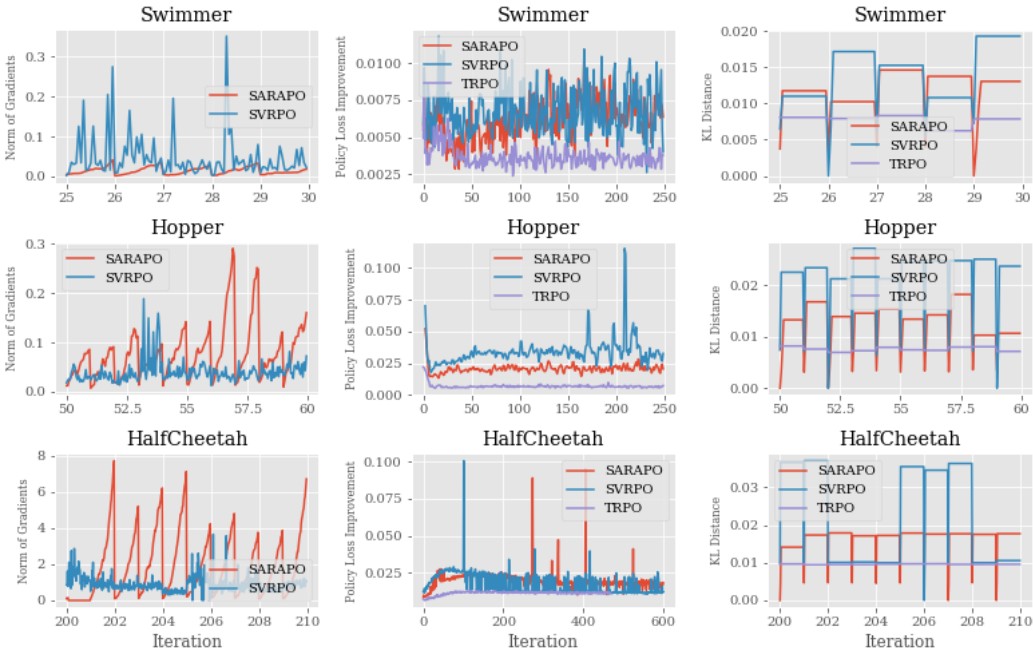

Figure 2: An illustration of the policy loss improvement, the estimated gradients norm and the KL divergence during training. The first row is the results on the Swimmer task. The second row is the results on the Hopper task. The third row is the results on the HalfCheetah. The pictures are zoomed in to show more details and the x-axis is normalized to be the number of outer loop iterations.

We zoom in the plot and choose the range of our x-axis to be in an interval that the mean episode reward is increasing. Note that this is a result of one single random run. The training curve of this run is shown in the Appendix.

In the first column, we plot the $L^2$ norm of the estimated gradient of SVRG and SARAH. Then the second column is the loss function both before and after an outer loop update. For SARAPO and SVRPO, this statistics is the sum of the increase in each small step update. We compare the result with the policy loss improvement in TRPO. From this, we verify our conjecture that both SARAH and SVRG reach a better optimum than SGD. Generally, a more significant increase in loss corresponds to a higher mean episode rewards.

The third column shows the KL divergence between the old policy and the updated policy. We observe that although we do not constrain the KL strictly in the line search, the KL of TRPO stays under the given threshold. In the interval to plot the result, KL constraint is set to be 0.01, TRPO stays under 0.01, SARAPO stays under 0.02, and SVRPO stays under 0.03. This is a good illustration that KL divergence won't step two far away after several inner loop updates. The variance reduced gradient finds a better solution slightly outside the KL range.

## 5    CONCLUSION

In this work, we propose a sample-efficient and variance reduction method for reinforcement learning called Stochastic Recursive Gradient Policy Optimization (SARAPO) algorithm. We hybrid the recently proposed SARAH algorithm which is a kind of variance reduction method with TRPO, which is the state-of-the-art model-free policy gradient methods in continuous control tasks together. We also provide theory which presents our algorithm enjoys the better properties by using ordinary and stochastic differential equations. Our experiments show its advantage over existing policy optimization methods such as TRPO and SVRPO. We hope this work can inspire future works and ultimately achieve higher performance in the reinforcement learning tasks.

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

## A HYPERPARAMETERS

Here, we present the hyper-parameters that get the best results for all tasks. We tune hyper-parameter by using grid search and record the hyper-parameter in Table 3 and Table 4. Specifically, the mini-batch size is searched in $\{1000, 5000\}$, the inner loop iterations is in $\{20, 50\}$, the max KL is between 0.005 and 0.01, and the Conjugate Gradient (CG) damping factor is 0.1.

| Tasks | Minibatch Size | Inner Loop Iteration | Max KL | CG Damping Factor |
|---|---|---|---|---|
| Pendulum | 5000 | 50 | 0.005 | 0.1 |
| CartPole | 5000 | 50 | 0.006 | 0.1 |
| Swimmer | 5000 | 20 | 0.005 | 0.1 |
| Half-Cheetch | 5000 | 50 | 0.005 | 0.1 |
| Walker | 1000 | 20 | 0.005 | 0.1 |
| Hopper | 1000 | 20 | 0.005 | 0.1 |

Table 3: The Best Hyper-Parameter of SARAPO

| Tasks | Minibatch Size | Inner Loop Iteration | Max KL | CG Damping Factor |
|---|---|---|---|---|
| Pendulum | 5000 | 50 | 0.01 | 0.1 |
| CartPole | 5000 | 50 | 0.01 | 0.1 |
| Swimmer | 1000 | 20 | 0.01 | 0.1 |
| Half-Cheetch | 5000 | 50 | 0.01 | 0.1 |
| Walker | 5000 | 50 | 0.01 | 0.1 |
| Hopper | 5000 | 20 | 0.01 | 0.1 |

Table 4: The Best Hyper-Parameter of SVRPO

## B THE MEAN EPISODE REWARD OF THE EXPERIMENTS IN FIGURE 4.3

We provided the learning curve for the experiments in Figure 4.3. We only plot the statistics using a single run. We see that for this random seed, SARAPO achieve best performance on Swimmer and HalfCheetah tasks. SARAPO and SVRPO get similar performance on Walk2d task and both outperform than TRPO except Hopper task.

## C EXPERIMENT ON AN ADAPTIVE PENALTY VARIANT OF THE PROXIMAL POLICY MAPPING

To find out more about applying stochastic gradient in policy gradient algorithms and its variants. We also proposed an algorithm on the Proximal Policy Gradient (PPO). We use the adaptive KL penalty version of PPO because the clipped objective function is not differentiable. We list our algorithm that combined SVRG and PPO in our appendix. And did experiments on Swimmer to compare the performance of SARAPPO and SVRPPO. Both SARAPPO and SVRPPO reaches a better reward than PPO.

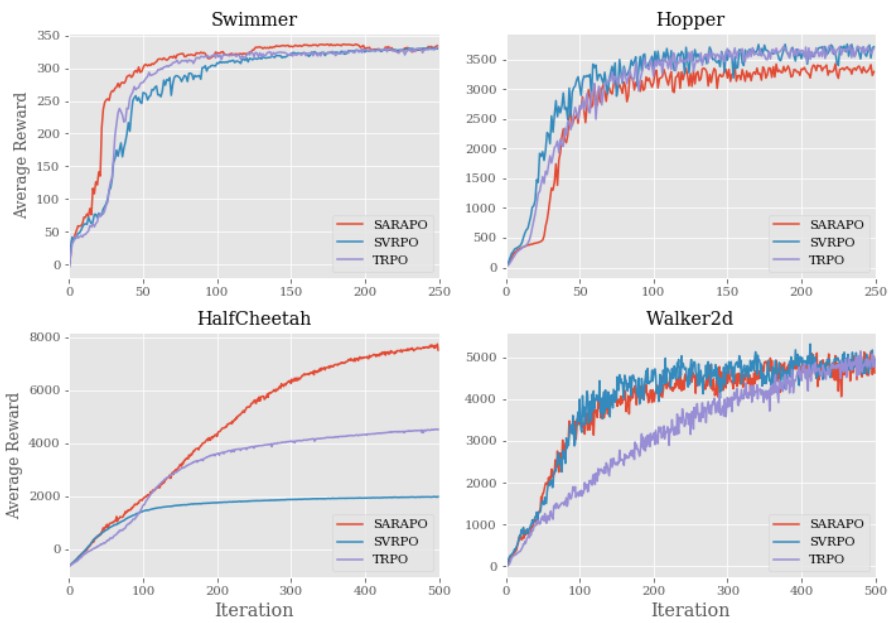

Figure 3: The Mean Episode Reward of the experiments in Figure 4.3

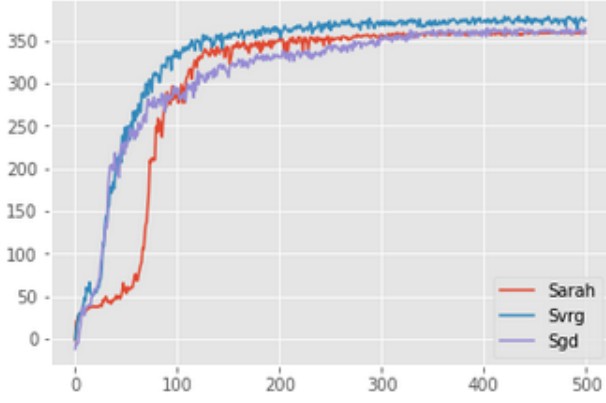

Figure 4: Comparison of Stochastic Recursive Proximal Policy Gradient to PPO

