# OpenReview forum: "Policy Optimization via Stochastic Recursive Gradient Algorithm"
_ICLR.cc/2019/Conference_

### Official Review · AnonReviewer1 · 2018-11-03
**Advantages of the proposed method over SVRG + policy gradient method are unclear.**

**Rating:** 5
**Confidence:** 3

**Review:**

This paper proposes a new policy gradient method for reinforcement learning.
The method essentially combines SARAH and trust region method using Fisher information matrix.
The effectiveness of the proposed method is verified in experiments.

SARAH is a variance reduction method developed in stochastic optimization literature, which significantly accelerates convergence speed of stochastic gradient descent.
Since the policy gradient often suffers from high variance during the training, a combination with variance reduction methods is quite reasonable.
However, this work seems to be rather incremental compared to a previous method adopting another variance reduction method (SVRG) [Xu+2017, Papini+2018].
Moreover, the advantage of the proposed method over SVRPG (SVRG + policy gradient) is unclear both theoretically and experimentally.
[Papini+2018] provided a convergence guarantee with its convergence rate, while this paper does not give such a result.
It would be nice if the authors could clarify theoretical advantages over SVRPG.

Minor comment:
- The description of SVRG updates in page 2 is wrong.
- The notation of H in Section 3.1 ("ODE analysis") is not defined at this time.

---

### Official Review · AnonReviewer3 · 2018-11-05
**Sarah applied to policy optimization with comparable performance to SVRG**

**Rating:** 6
**Confidence:** 2

**Review:**

The paper extends Sarah to policy optimization with theoretical analysis and experimental study.

1) The theoretical analysis under certain assumption seems novel. But the significance is unknown compared to similar analysis.

2) The analysis demonstrates the advantage of Sarah over SVRG, as noted in Remark 1. It would be better to give explicit equations for SVRG in order for comparison.

3) Experimental results seem to show empirically that the SARAH is only comparable to SVRG.

4) Presentation needs to be improved.

---

### Official Review · AnonReviewer4 · 2018-11-26
**TRPO with SARAH optimization: great idea, but inconclusive results**

**Rating:** 5
**Confidence:** 3

**Review:**

This paper investigates how the SARAH stochastic recursive gradient algorithm can be applied to Trust Region Policy Optimization. The authors analyze the SARAH algorithm using its approximating ordinary and stochastic differential equations. The empirical performance of SARAPO is then compared with SVRPO and TRPO on several benchmark problems.

Although the idea of applying SARAH to reduce the variance of gradient estimates in policy gradient algorithms is interesting and potentially quite significant (variance of gradient estimates is a major problem in policy gradient algorithms), I recommend rejecting this paper at the present time due to issues with clarity and quality, particularly of the experiments.

Not enough of the possible values for experimental settings were tested to say anything conclusive about the performance of the algorithms being compared. For the values that were tested, no measures of the variability of performance or statistical significance of the results were given. This is important because the performance of the algorithms is similar on many of the environments, and it is important to know if the improved performance of SARAPO observed on some of the environments is statistically significant or simply due to the small sample size.

The paper also needs improvements in clarity. Grammatical errors and sentence fragments make it challenging to understand at times. Section 2.3 seemed very brief, and did not include enough discussion of design decisions made in the algorithm. For example, the authors say ``"the Fisher Information Matrix can be approximated by Hessian matrix of the KL divergence when the current distribution exactly matches that of the base distribution" but then suggest using the Hessian of the KL of the old parameters and the new parameters which are not the same. What are the consequences of this approximation? Are there alternative approaches?

The analysis in section 3 is interesting, but the technique has been applied to SGD before and the results only seem to confirm findings from the original SARAH paper.

To improve the paper, I would suggest moving section 3 to an appendix and using the extra space to further explain details and conduct additional simpler experiments. Additional experiments on simpler environments and policy gradient algorithms (REINFORCE, REINFORCE with baseline) would allow the authors to try more possible values for experimental settings and do enough runs to obtain more conclusive results about performance. Then the authors can present their results applying SARAH to TRPO with some measure of statistical significance.

---

### Author Response · Authors · 2018-11-26
**Respond to all reviewers**

We sincerely thank all reviewers for the valuable remarks!

We would like to emphasize that our paper is not an incremental one. We believe that variance reduced (VR) gradient methods (SARAH and SVRG) serve as potential alternatives to incorporate into the TRPO framework [Xu 2017], which might significantly outperform the PG-type algorithms accompanied by importance sampling [Papini 2018]. We aimed to provide the first theoretical analysis in order to support the experiments of VR gradient methods [Xu 2017], and the differential equation approximation for VR is a novel and powerful tool to analyze such.

Despite saying that, we agree that our experiments might not be sufficient to convince some of our proposal. This is partly due to the limited time for running large-scale experiments. Following reviewers' remarks, will try to work more smaller test experiments to support our proposal and fix all the clarity/presentation issues and typos in our next submission.

---

### Meta-Review · Area_Chair1 · 2018-12-13
**Interesting idea that needs a bit more work**

**Confidence:** 4
**Recommendation:** Reject

**Metareview:**

The use of SARAH for Policy optimization in RL is novel, with some theoretical analysis to demonstrate convergence of this approach. However, concerns were raised in terms of clarity of the paper, empirical results and in placement of this theory relative to a previous variance reduction algorithm called SVRPG. The author response similarly did not explain the novelty of the theory beyond the convergence results of what was given by the paper on SVRPG.  By incorporating some of the reviewer comments, this paper could be a meaningful and useful contribution.